# VerFormer: Vertebrae-Aware Transformer for Automatic Spine Segmentation from CT Images

**DOI:** 10.3390/diagnostics14171859

**Published:** 2024-08-25

**Authors:** Xinchen Li, Yuan Hong, Yang Xu, Mu Hu

**Affiliations:** Department of Orthopedics, Ruijin Hospital, Shanghai Jiao Tong University School of Medicine, Shanghai 200025, China; leexingchen2011@163.com (X.L.); suppowerhong@163.com (Y.H.); dr_xuyang@163.com (Y.X.)

**Keywords:** Vision Transformer, spine CT segmentation, attention mechanism

## Abstract

The accurate and efficient segmentation of the spine is important in the diagnosis and treatment of spine malfunctions and fractures. However, it is still challenging because of large inter-vertebra variations in shape and cross-image localization of the spine. In previous methods, convolutional neural networks (CNNs) have been widely applied as a vision backbone to tackle this task. However, these methods are challenged in utilizing the global contextual information across the whole image for accurate spine segmentation because of the inherent locality of the convolution operation. Compared with CNNs, the Vision Transformer (ViT) has been proposed as another vision backbone with a high capacity to capture global contextual information. However, when the ViT is employed for spine segmentation, it treats all input tokens equally, including vertebrae-related tokens and non-vertebrae-related tokens. Additionally, it lacks the capability to locate regions of interest, thus lowering the accuracy of spine segmentation. To address this limitation, we propose a novel Vertebrae-aware Vision Transformer (VerFormer) for automatic spine segmentation from CT images. Our VerFormer is designed by incorporating a novel Vertebrae-aware Global (VG) block into the ViT backbone. In the VG block, the vertebrae-related global contextual information is extracted by a Vertebrae-aware Global Query (VGQ) module. Then, this information is incorporated into query tokens to highlight vertebrae-related tokens in the multi-head self-attention module. Thus, this VG block can leverage global contextual information to effectively and efficiently locate spines across the whole input, thus improving the segmentation accuracy of VerFormer. Driven by this design, the VerFormer demonstrates a solid capacity to capture more discriminative dependencies and vertebrae-related context in automatic spine segmentation. The experimental results on two spine CT segmentation tasks demonstrate the effectiveness of our VG block and the superiority of our VerFormer in spine segmentation. Compared with other popular CNN- or ViT-based segmentation models, our VerFormer shows superior segmentation accuracy and generalization.

## 1. Introduction

Diagnosing and treating pathological diseases demands accurate spine segmentation and vertebrae identification. Vertebral segmentation and identification play a vital role in supporting spine-related clinical workflow, including diagnosing vertebral and spinal deformities and computer-assisted surgical planning [1]. However, manual segmentation by surgeons and radiologists is labor-intensive and time-consuming, motivating the development of automatic and semi-automatic tools [2]. The intricate nature of spine anatomy, characterized by its complex geometry and varying tissue intensities, poses significant challenges to traditional automatic segmentation techniques. However, the advent of deep learning (DL) algorithms has revolutionized the accuracy and efficiency of spine segmentation and vertebrae identification tasks [3,4]. Among various DL-based methods, convolutional neural networks (CNNs) have been widely used, and these CNN-based methods are performed for automatic spine segmentation and vertebrae identification in various medical image modalities, such as Computed Tomography (CT) [5], Magnetic Resonance Imaging (MRI) [6], and X-ray imaging [7]. However, spine segmentation by CNNs from CT images is still challenging. The spine has a complex anatomical structure and variability in the shape and size of the vertebrae. Additionally, the spine is always located across the whole CT image. However, due to the intrinsic characteristics of convolutional operations, CNNs are limited in utilizing image-level global contextual information to capture vertebrae-aware features. Thus, this lowers the capabilities of CNNs on accurate segmentation of the spine from CT images.

Unlike CNNs, the Vision Transformer (ViT) can capture and utilize image-level contextual information [8]. The significant factor in the success of ViT-based models is the application of the self-attention module, which enables them to utilize large receptive fields to capture global contextual information across the entire input image. Thus, this benefits ViTs in serving as a vision backbone for vertebrae segmentation from CT images, but few works are proposing ViT-based methods for vertebrae segmentation. To demonstrate the potential power of ViTs on vertebra and spine segmentation, we propose a ViT-based segmentation method. However, the standard ViT partitions the whole images into tokens and treats these tokens equally as the query in multi-head self-attention modules [8]. Thus, the ViT lacks the capabilities of highlighting vertebrae-related tokens and locating the region of interest in the spine from CT images, leading to the mis-segmentation or under-segmentation of the whole spine.

To tackle this limitation, we propose a novel and efficient ViT-based model for automatic spin segmentation from CT images, termed Vertebrae-aware Vision Transformer (VerFormer). Our method is designed based on employing a novel Vertebrae-aware Global (VG) block to efficiently and effectively capture vertebrae-related contextual information and facilitate the ViT to locate the spine from the whole input image, thus improving segmentation performance. This is achieved in the VG block by incorporating a novel Vertebrae-aware Global Query (VGQ) module. This VGQ module can capture vertebrae-aware contextual information with attention mechanisms and inject this information into query tokens to highlight vertebrae-related tokens based on their attention scores. When these tokens are encoded in the multi-head self-attention module, vertebrae-related tokens with higher attention scores are emphasized, and the spine is located. This mechanism can improve the capabilities of our VerFormer on spine segmentation from CT images. We evaluate our VerFormer for spine segmentation from CT images on two widely used public datasets. Our proposed model achieves superior segmentation performance in spine segmentation compared to other state-of-the-art methods on these two benchmarks. Our contributions can be summarized as follows.

–We propose a novel Vertebrae-aware Vision Transformer (VerFormer) for automatic spine segmentation from CT images. It employs the Vision Transformer as the backbone to utilize image-level global contextual information for accurate spine segmentation.–We propose a novel Vertebrae-aware Global (VG) block to efficiently and effectively capture vertebrae-related contextual information and facilitate the ViT to locate the spine from the whole input image, thus improving segmentation performance.–We incorporate a novel Vertebrae-aware Global Query (VGQ) module into the Vertebrae-aware Global (VG) block. This module can highlight vertebrae-related query tokens based on their attention scores when they are encoded in the multi-head self-attention module. Emphasizing vertebrae-related tokens empowers our VerFormer to locate the spine for accurate segmentation.–We evaluate our VerFormer on two widely used spine segmentation datasets, VerSe 2019 and 2022. It achieves superior performance than other state-of-the-art methods.

## 2. Related Work

### 2.1. Vertebra Segmentation by Convolutional Neural Networks

Before deep learning was widely used in medical image analysis, vertebra segmentation was mainly solved by statistical shape models, including active shape models and shape-constrained deformable models [9,10,11,12,13,14]. In addition, some other traditional methods were proposed for vertebra segmentation, such as atlas-based models [15], level-sets with shape priors [16], and active contours [17,18].

Recently, deep learning-based models have been proposed for vertebra and spine segmentation. Korez et al. used a convolutional neural network (CNN) to generate probability maps for the vertebra regions and then used these maps to generate a deformable statistical model to segment the spine [3]. Sekuboyina et al. used a multi-class CNN to generate labels for the lumbar vertebrae in 2D spine CT slices [19]. Subsequently, these labels were used as prior information, and a multi-layer perception (MLP) was applied to generate a bounding box for the lumbar region and to identify the region of interest from the images. In subsequent work, Sekuboyina et al. designed a 3D patch-based CNN for vertebra segmentation by performing voxel classification from the whole image [4]. To further boost the performance by removing false positives outside the region of interest, a 2D network was used to predict a low-resolution mask for the vertebral region. Lessmann et al. applied a two-stage iterative approach for accurate vertebra segmentation [20]. Specifically, a CNN was used to segment vertebrae in downsampled images iteratively. Then, another CNN was used to analyze the full-resolution images to refine the low-resolution segmentation results. He et al. presented a dual densely connected U-shaped network (DDU-Net) to perform the extraction of multi-scale features and segmentation of different sizes of tissues automatically and precisely for automatic vertebral segmentation in CT images [21]. Liping et al. proposed a lightweight pyramid attention quick refinement network (LPAQR-Net) for efficient and accurate vertebra segmentation from biplanar whole spine radiographs [22].

### 2.2. Vision Transformer for Medical Image Segmentation

The recently proposed Vision Transformer (ViT) shows great success in computer vision tasks by utilizing a self-attention mechanism to capture long-range dependencies [8]. Because of its promising performance, ViT has been used as a backbone for medical image segmentation. TransUNet was the first ViT-based medical image segmentation model, which combines a ViT encoder with a CNN decoder [23]. Swin UNet was proposed as a pure Transformer-based model for medical image segmentation by incorporating the Swin Transformer block into a hierarchical architecture [24]. Subsequently, UNETR combined a Transformer backbone and a CNN architecture for 3D volumetric segmentation [25]. Similarly, Swin UNETR utilized the Swin Transformer in the encoder for brain tumor segmentation [26]. Following the idea of nnUNet [27], nnFormer was a self-configured ViT-based segmentation model for brain tumor segmentation [28]. CoTr incorporated a Deformable Transformer into a CNN architecture for accurate segmentation [29]. Hiformer was designed as a hierarchical multi-scale ViT-based model for medical image segmentation [30]. AgileFormer was proposed to capture spatially deformable features to improve the performance of medical image segmentation [31]. MS-Former employed a dual-branch Transformer network to encode global contextual dependencies while preserving local information by extracting two different scaled features [32].

### 2.3. Channel and Spatial Attention

Many methods and modules have been proposed to calculate channel attention to facilitate networks to utilize long-range dependencies. Squeeze-and-Excitation was proposed to model channel-wise inter-dependencies and use these channel attention values to recalibrate channel-wise features [33]. Then, another more efficient channel attention module was proposed to calculate channel attention in an efficient manner [34]. Channel attention is calculated in these independent modules. However, it is also calculated to improve convolution layers in CNNs and local-window-based self-attention in hierarchical ViTs. The channel-wise inter-dependencies among features from two convolution layers were modeled in selective kernel networks [35] and channel attention networks [36]. Similarly, channel attention is calculated to model inter-dependencies among several convolutional layers [37]. Several methods have also been proposed to calculate spatial attention to enhance the capabilities of networks to model spatial-wise global contextual information. Specifically, spatial-wise attention was modeled in the CBAM [38] and dual attention networks [39]. The spatial-wise inter-dependencies among features from two convolution layers were modeled in spatial dynamic networks [40]. The attention gate was also designed to enable networks to suppress irrelevant regions and highlight salient spatial features by calculating spatial-wise attention values [41].

## 3. Methods

In this section, we first introduce the overall architecture of our Vertebrae-aware Vision Transformer (VerFormer) in Section 3.1. Then, we demonstrate the architecture of the Vertebrae-aware Global block, which is the basic block of the VerFormer in Section 3.2. This Vertebrae-aware Global block is built by incorporating the novel Vertebrae-aware Global Query module into the multi-head self-attention module. In Section 3.3, we introduce the loss function used for model training.

### 3.1. Overall Network

The overall architecture of the proposed Vertebrae-aware Vision Transformer (VerFormer) is demonstrated in Figure 1. The VerFormer is built as a U-shaped encoder-decoder architecture and consists of an encoder, a bottleneck, and a decoder. The overall architecture is designed based on a hierarchical Vision Transformer framework to obtain multi-scale feature representations. The basic block of our VerFormer is the Vertebrae-aware Global (VG) block, and two successive blocks are incorporated into each stage. The basic number of channels is C=96, and the numbers of channels in the following stages are 192, 384, and 768. The number of channels in the bottleneck is 768. To extract more features in the deeper layers, the number of heads in the multi-head self-attention module at each stage is 2, 4, 8, and 16, respectively.

In the encoder, given an input image patch with the resolution H×W×1, we obtain overlapping patches with the resolution $\frac{H}{2}\times \frac{W}{2}\times 1$ by applying a 3×3 convolutional layer with a stride of 2. Subsequently, these patches are projected onto a *C*-dimensional embedding space by another 3×3 convolutional layer with stride 2. In this convolutional layer, the dimension of these patches is $\frac{H}{4}\times \frac{W}{4}\times C$. The encoder consists of three stages. At each stage, two successive VG blocks are applied to extract spatial features. Subsequently, a downsampling layer is applied to decrease the spatial resolution by 2 while increasing the number of channels by 2 via a 3×3 convolutional layer with a stride of 2. The dimensions of patches in three stage are $\frac{H}{4}\times \frac{W}{4}\times C$, $\frac{H}{8}\times \frac{W}{8}\times 2C$, and $\frac{H}{16}\times \frac{W}{16}\times 4C$, respectively.

The bottleneck employs two successive VG blocks, which are used to learn the deep feature representations. In the bottleneck, the dimension and resolution of input and output features are kept unchanged as $\frac{H}{32}\times \frac{W}{32}\times 8C$.

We design a symmetric decoder as the encoder for dense prediction. The decoder also consists of three stages. At each stage in the decoder, we first apply an upsampling layer to increase the spatial resolution and decrease the channel number by 2 via a 2×2 transposed convolutional layer with a stride of 2. The dimension of patches is $\frac{H}{16}\times \frac{W}{16}\times 4C$. Then, the upsampled features are concatenated with features from the encoder via skip connections. Subsequently, a 1×1 convolutional layer is used to decrease the number of channels by 2 as $\frac{H}{16}\times \frac{W}{16}\times 4C$. In this stage, two successive VG blocks are applied to extract spatial features. The dimensions of patches in three stages are $\frac{H}{16}\times \frac{W}{16}\times 4C$, $\frac{H}{8}\times \frac{W}{8}\times 2C$, and $\frac{H}{4}\times \frac{W}{4}\times C$, respectively. To recover the feature to the original dimension of H×W×C, a final projection layer consisting of two transposed convolution layers with the kernel of 2×2 and a stride of 2 is used. Finally, a 1×1 convolutional layer is applied to produce the pixel-wise segmentation prediction.

### 3.2. Vertebrae-Aware Global Block

Figure 2 demonstrates the architecture of our Vertebrae-aware Global (VG) block. Our VG block is the basic block that is used in our VerFormer to capture vertebrae-aware global contextual features. The multi-head self-attention module is followed by a 2-layer MLP module with GELU non-linearity in between. A Layer Normalization (LN) layer is applied before each self-attention module and each MLP module, and a residual connection is applied after each module.

#### 3.2.1. Vertebrae-Aware Global Query Module

Each VG block employs a Vertebrae-aware Global Query (VGQ) module to extract vertebrae-aware contextual information based on attention mechanisms. Then, this attention information can be utilized to highlight vertebrae-related tokens and suppress non-vertebrae-related tokens. Figure 2 demonstrates the architecture of our Vertebrae-aware Global Query (VGQ) module. Figure 3 demonstrates the mechanism of our Vertebrae-aware Global Query (VGQ) module. In our VGQ module, a channel attention-based module and a spatial attention-based module are both used to extract vertebra-related contextual information. This channel attention-based module cascades a 3×3 depth-wise convolutional layer (DWConv), a GELU activation function, a Squeeze-and-Excitation (SE) module [33], and a 1×1 convolutional layer (Conv). This SE module consists of an average pooling (AvgPool) layer and two fully connected (FC) layers with a ReLU activation function in between. This channel attention-based module can be formulated as Equation (Equation 1).
(1)$$\begin{array}{cc} \hat{x} & =\mathrm{DWConv}(x),\\ \hat{x} & =\mathrm{SE}(\mathrm{GELU}\left(\hat{x}\right)),\\ \hat{x} & =\mathrm{Conv}(\hat{x}) \end{array}$$

The spatial attention-based module cascades a 3×3 depth-wise convolutional layer (DWConv), a GELU activation function, an adaptive pooling (AvgPool) layer, a 7×7 convolutional layer (${\mathrm{Conv}}_{7}$), and a 1×1 convolutional layer (Conv). This channel attention-based module can be formulated as Equation (Equation 2).
(2)$$\begin{array}{cc} \hat{x} & =\mathrm{DWConv}(x),\\ \hat{x} & ={\mathrm{Conv}}_{7}\left(\mathrm{AvgPool}\left(\mathrm{GELU}\left(\hat{x}\right)\right)\right),\\ \hat{x} & =\mathrm{Conv}(\hat{x}) \end{array}$$

Features extracted from the channel attention-based module and spatial attention-based module are combined and then reshaped into tokenized features with the dimension (h∗w)×C. Subsequently, these are repeated and transformed to generate vertebrae-aware global queries $q_{v}$ with the dimension of B×(h∗w)×C. This output of our VGQ module is fed into the multi-head self-attention module.

#### 3.2.2. Multi-Head Self-Attention Module

Unlike local self-attention in the Swin Transformer, which can only query patches within a local window [42], our multi-head self-attention (MHSA) module is designed to query vertebrae-related image regions. These vertebrae-related query tokens are generated by capturing global channel attention and spatial attention information in the VGQ module. Specifically, at each stage before the self-attention module, the VGQ module is used to pre-compute Vertebrae-aware Global Query tokens $q_{v}$ based on global information. Then, vertebrae-related tokens are highlighted by their attention scores. The multi-head self-attention module utilizes the extracted vertebrae-aware global queries to interact with the key and value representations. With this design, our multi-head self-attention module can effectively utilize global information and capture spatial features. Theoretically, the Vertebrae-aware Global Query $q_{v}$ has a size of B×C×h×w, where *B*, *C*, *h*, and *w* denote batch size, embedding dimension, local window height, and width, respectively. $q_{v}$ is further reshaped and fed into multi-heads in the self-attention modules, and the value *v* and key *k* are then computed using a linear layer. Since the partitioned windows only contain local information [42], utilizing rich global contextual information enlarges the receptive field of the network. Additionally, generating vertebrae-aware global queries provides an effective way of extracting localization information and related regions of vertebrae in the input feature maps. The multi-head self-attention module is computed in Equation (Equation 3).
(3)$$\begin{array}{c} Attention\left(q_{v},k,v\right)=softmax\left(\frac{q_{v}k}{\sqrt{d}}+b\right)v \end{array}$$
where *d* is a scaling factor and *b* is a learnable relative position bias term.

### 3.3. Loss Function

We employ a combo loss function (Equation (Equation 4)), which is a combination of dice loss and cross-entropy loss.
(4)$$\begin{array}{c} L=\lambda_{1}L_{DICE}+\lambda_{2}L_{CE} \end{array}$$

The optimal values for $\lambda_{1}$ and $\lambda_{2}$ are 0.6 and 0.4, respectively. Dice loss [43] is employed to measure the dissimilarity between the predicted segmentation and the group truth segmentation of the targeting object (Equation (Equation 5)).
(5)$$\begin{array}{c} L_{DICE}=\frac{2\times P\times Y+\epsilon }{P+Y+\epsilon } \end{array}$$

Cross entropy loss [44] is employed to measure the error between two probability distributions of predicted segmentation of the targeting object (Equation (Equation 6)).
(6)$$\begin{array}{c} L_{CE}=-\frac{1}{N}\left[\sum_{i=1}^{N}\left[p_{i}\log\left(p_{i}\right)+\left(1-p_{i}\right)\log\left(1-p_{i}\right)\right]\right] \end{array}$$

## 4. Results

### 4.1. Datasets

To validate our segmentation methods, we used two publicly available datasets for automatic vertebra and spine segmentation from CT images, including VerSe 2019 and VerSe 2020 [45,46,47]. These two datasets were prepared as vertebral labeling and segmentation challenges hosted at the 2019 MICCAI and 2020 MICCAI, respectively. Both datasets can be used by researchers to evaluate their segmentation models for the segmentation of the spine with multiple conditions, labeled vertebrae, and field of view from CT images. Specifically, the VerSe 2019 dataset includes 160 CT scans from 141 patients with metallic implants or spinal fractures, as well as a combination of isotropic and sagittal reformations. Centroids and segmented masks are provided for all scans, and these segmentation masks are used as ground truth in our experiment. The VerSe 2020 dataset consists of 300 CT scans with manually annotated voxel-wise labels. Scans from this dataset were collected across multiple centers from four different scanner manufacturers. This dataset was enriched with cases that exhibit anatomical variants, such as enumeration abnormalities and transitional vertebrae.

We implemented several pre-processing techniques for the data from two datasets. Three-dimensional scans were resampled to the target spacing of 1×1×1 mm. Subsequently, voxel intensities were normalized by implementing the max–min normalization technique after being truncated by the percentage [5%,95%]. Finally, normalized 3D volumes were prepared into 2D patches with the patch size of 224×224. Before training, several data augmentation techniques were applied. Specifically, patches were rotated with [−30,30] and scaled with [0.8,1.2] with both a probability of 0.3. Then, patches were mirrored along the X and Y axes with a probability of 0.5. Gaussian noise with a mean of zero and a variance of 0.1 was added to these patches with a probability of 0.2. Contrasts and brightness were added to pixel intensities in patches with a probability of 0.1.

### 4.2. Implementation Details

Our VerFormer was implemented based on PyTorch 3.9 and CUDA 11.3. During the training experiments, we used stochastic gradient descent (SGD) as the optimizer with a momentum of 0.99. Models were trained for 500 epochs by an initial learning rate of 0.005 with a poly weight decay factor of $3\times 10^{-5}$. In all experiments, we adopted a batch size of 16. We trained all models on a single NVIDIA GeForce RTX 3090 GPU from the server of Shanghai Jiao Tong University (Shanghai, China) with 24 GB of memory. Similar to other ViT-based segmentation networks [24,26,31], we used the input patches with a resolution of 224×224. Five-fold cross-validation was implemented for evaluation to avoid overfitting and to provide a more accurate estimate of the model’s generalization performance.

### 4.3. Evaluation Metrics

Five evaluation metrics were used to validate the segmentation performance of our proposed model. The dice similarity coefficient (DSC) is the most common and useful evaluation metric for segmentation tasks. It measures the similarity between the prediction mask and ground truth. The mathematical formula for calculating the DSC is shown as Equation (Equation 7), where *P* and *Y* are the predicted image and the ground truth, respectively.
(7)$$\mathrm{DSC}=\frac{2\times (P\times Y)}{P+Y}$$

Intersection over union (IoU) is also used to evaluate the performance of our automatic segmentation methods. It is used to compare a predicted mask with a known mask for semantic segmentation. The mathematical formula for the IoU is shown as Equation (Equation 8).
(8)$$\mathrm{IoU}=\frac{TP}{TP+FN+FP}$$

Precision is calculated by quantifying the total number of correct positive outcomes made by the proposed model. The mathematical formula for calculating the precision is shown as Equation (Equation 9).
(9)$$\mathrm{Precision}=\frac{TP}{TP+FP}$$

The recall is calculated as the total true positive divided by the sum of the true positive and false negative. The mathematical formula for the recall is shown as Equation (Equation 10).
(10)$$\mathrm{Recall}=\frac{TP}{TP+FN}$$

The 95% percentile Hausdroff Distance (95HD) is used to measure the maximum distance of the prediction set to the nearest point in the ground truth set. The mathematical formula for the 95HD is shown as Equation (Equation 11).
(11)$$95\mathrm{HD}=\max(\max_{x\in X}\min_{y\in Y}||x-y||,\max_{y\in Y}\min_{x\in X}||y-x||).$$

### 4.4. Experimental Results

We compared our VerFormer with 12 state-of-the-art (SOTA) methods. Four methods are CNN-based architecture, including U-Net [48], V-Net [49], Attention UNet (Att-UNet) [50], and nnU-Net [27]. Five methods are hybrid CNN and Transformer architecture, including TransUNet [23], UCTransNet [51], 3D UNETR [25], Swin UNETR [26], and 3D UX Net [52]. Among these methods, UNETR, Swin UNETR, and UX Net are 3D models. Three methods employ pure Vision Transformer as backbones, including Swin UNet [24], MissFormer [53], and AgileFormer-B [31]. We used the default configurations of these models provided in their papers.

Table 1 demonstrates the comparison results of the segmentation performance of our methods and other methods on the VerSe 2019 dataset. In comparison with other vertebra segmentation methods, our VerFormer outperformed previous methods on spine segmentation. Specifically, when compared with widely used 2D CNN baselines, U-Net and VNet, our VerFormer exceeded in performance by over 5% in DSC, IoU, and precision; 6% in recall; and six points in 95HD. Additionally, our VerFormer showed a better segmentation performance than 3D CNN-based models, including Att-UNet and nnU-Net, even though our VerFormer was evaluated as a 2D segmentation method. Moreover, our VerFormer showed a higher segmentation accuracy than other 2D or 3D CNN-Transformer methods with over 1–6% improvements in DSC, IoU, precision, recall, and 95HD points. Finally, our VerFormer had a much higher segmentation accuracy than other Vision Transformer-based segmentation methods, including Swin UNet, MissFormer, and AgileFormer.

Table 2 demonstrates the comparison results of the segmentation performance of our methods and other methods on the VerSe 2020 dataset. Similar to the results in the VerSe 2019 dataset, our VerFormer outperformed these baseline methods on spine segmentation from CT images. Specifically, when compared with widely used 2D CNN baselines, U-Net and V-Net, our VerFormer still exceeded in performance by over 4% in DSC, 5% in IoU, 4% in precision points, 5% in recall points, and five points in 95HD. Additionally, our VerFormer also showed a better segmentation performance than 3D CNN-based models, including Att-UNet and nnU-Net (+2.5 DSC, +2.5 IoU, +3.5 precision, +4 recall, and +4 95HD), even though our VerFormer was evaluated as a 2D segmentation method. Moreover, our VerFormerr showed a higher segmentation accuracy than all CNN-Transformers methods, including TransUNet, UCTransNet, 3D UNETR, Swin UNETR, and 3D UX Net. Finally, our VerFormer had a much higher segmentation accuracy than other Vision Transformer-based segmentation methods, including Swin UNet, MissFormer, and AgileFormer.

## 5. Ablation Study

In the ablation study, we evaluated how the Vertebrea-aware Global (VG) block improves spine segmentation accuracy over the standard Vision Transformer block. It was implemented by comparing our VerFormerr with the standard ViT. Specifically, the standard ViT was built based on the standard Vision Transformer block by employing the multi-head self-attention (MHSA) module. Thus, we implemented the ablation study by deconstructing VerFormer by removing the VGQ module from the VG block.

The performance of our VerFormer and ViT was compared on two datasets, VerSe 2019 and VerSe 2020, and the results are shown in Table 3. In both datasets, our VerFormer demonstrated much better segmentation performance than the ViT. In these two datasets, our VerFormer outperformed the ViT by over four points in DSC, IoU, precision, and 95HD and five points in recall. Thus, our VerFormer improves the ViT on spine segmentation from CT images by incorporating VG block into it and utilizing the VGQ module to capture vertebrae-aware contextual information for spine localization. Figure 4 demonstrates the segmentation results from our VerFormer and global query tokens from the VGQ module.

## 6. Discussion

Our VerFormer was developed as a 2D ViT-based segmentation model for lower computational complexity. Our VerFormer cannot be developed as a 3D network because of the high computational complexity of multi-head self-attention. Our VerFormer generates tokens from the whole input image to capture image-level global contextual information. Then, features are extracted from these tokens in the multi-head self-attention (MHSA) module. If our VerFormer is designed as a 3D network, 3D tokens will be utilized in the MHSA module, thus leading to high computational complexity and high memory computation. Other 3D ViT-based networks, such as the Swin Transformer [26,42], utilize the local window and create 3D tokens within the window to avoid high computational complexity. However, employment of the local window limits these networks from capturing global information, thus lowering segmentation performance. Additionally, although our VerFormer is designed as a 2D model, it has achieved superior segmentation performance than other 3D methods, such as Att-UNet [41], 3D UNETR [25], Swin UNETR [26], and 3D UX Net [52].

Our VerFormer has high generalizability. It was evaluated on two datasets for spine segmentation from CT images. It can also be employed to spine CT images from other datasets since it is designed to extract data-driven features and can be trained end-to-end. Our VerFormer can also generalize across various medical image modalities. Our VerFormer can be applied to segment organs or lesions from 3D medical image modalities, such as CT and MRI. Additionally, it can also be used for other 2D medical images, such as X-rays and microscopy images. Thus, this 2D configuration improves the generalizability of our VerFormer over various medical image modalities.

The potential limitation of our VerFormer originates from the intrinsic characteristics of the ViT backbone. ViT extracts tokens of a fixed size from input images, and the VGQ module highlights vertebrae-related tokens. Thus, these fixed-sized tokens may limit our VerFormer from learning multi-scale features, and the size of the tokens may influence the segmentation performance. In future work, VerFormer can be improved by employing a dual path that extracts tokens of different sizes. Additionally, VerFormer can also be improved by utilizing an adaptive mechanism to find the optimal size of tokens based on input images.

## 7. Conclusions

In this paper, we proposed a Vertebrea-aware Vision Transformer, VerFormer, for automatic spine segmentation from CT images. Our VerFormer is designed by incorporating a novel Vertebrae-aware Global (VG) block into the ViT backbone. In the VG block, the vertebrae-related global contextual information is extracted by a Vertebrae-aware Global Query (VGQ) module. Then, this information is incorporated into query tokens to highlight vertebrae-related tokens in the multi-head self-attention module. Thus, this VG block can leverage global contextual information to effectively and efficiently locate spines across the whole input, thus improving the segmentation accuracy of VerFormer. We evaluate our VerFormer on two spine CT segmentation tasks, and experimental results demonstrate its superiority over other state-of-the-art methods. 

## Figures and Tables

**Figure 1 diagnostics-14-01859-f001:**
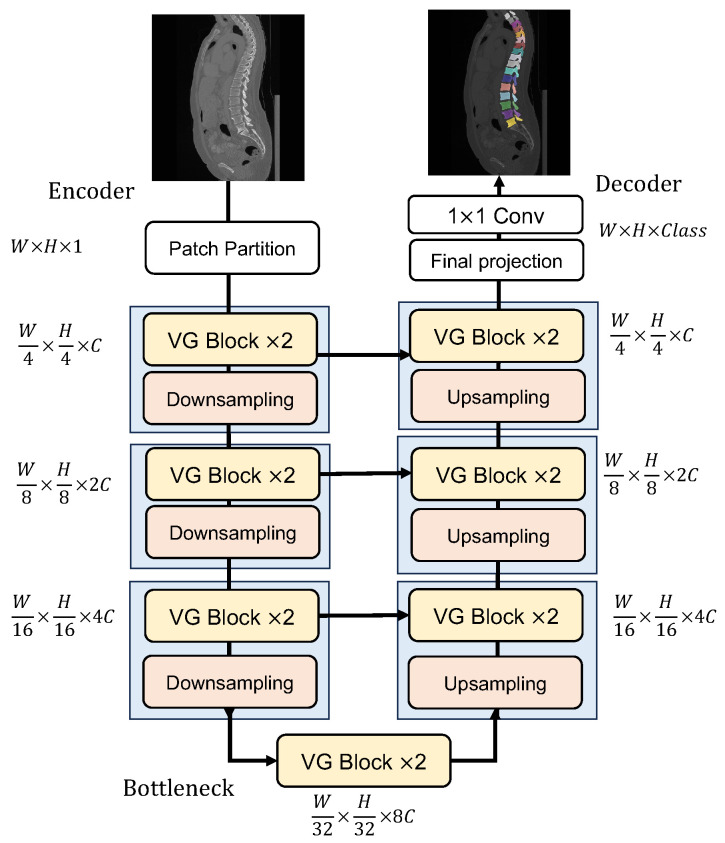
The overall architecture of the VerFormer. It consists of an encoder, a bottleneck, and a decoder. Both the encoder and the decoder consist of three stages, and in each stage, two successive VG blocks are used for learning feature representations. Two successive VG blocks are used in the bottleneck. A 3×3 convolutional layer with a stride of 2 is used for downsampling in each stage, and a 2×2 transposed convolutional layer with a stride of 2 is used for upsampling in each stage. Two 3×3 convolutional layers with a stride of 2 are used for the patch partition. Two 2×2 transposed convolution layers with a stride of 2 are used in the final projection.

**Figure 2 diagnostics-14-01859-f002:**
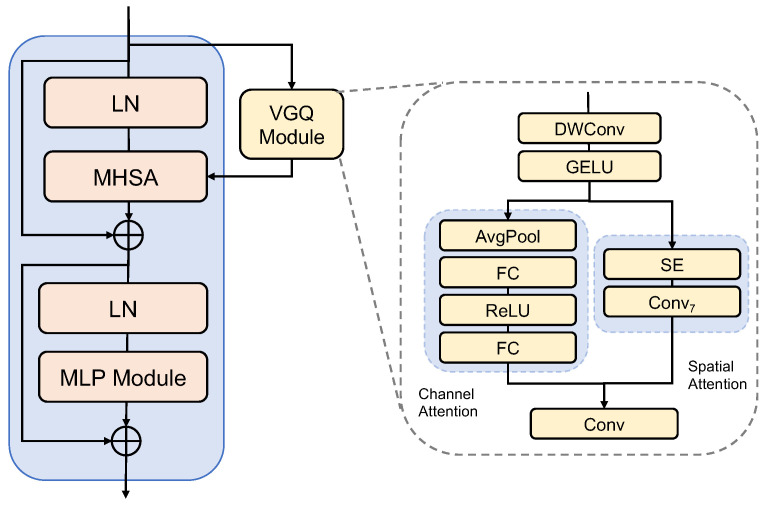
The overall architecture of the Vertebrae-aware Global (VG) block. In the VG block, a VGQ module is utilized to extract vertebrae-aware contextual information to highlight vertebrae-related query tokens. Then, these tokens are encoded in the multi-head self-attention (MHSA) module. In the VGQ module, two parallel paths are employed to extract global vertebrae-aware contextual information, including a channel attention path and a spatial attention path. In the channel attention path, an SE layer is used by cascading an average pooling (AvgPool) layer, a fully connected (FC) layer, a ReLu activation function, and a fully connected (FC) layer.

**Figure 3 diagnostics-14-01859-f003:**
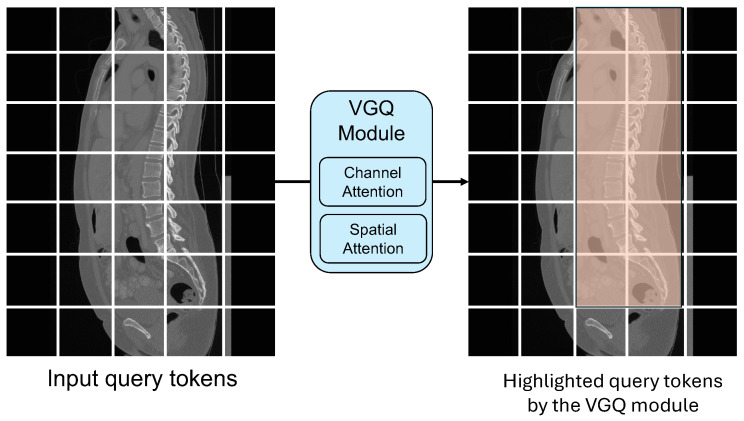
The mechanism of the Vertebrae-aware Global Query (VGQ) module. The input image is partitioned into patches, and these patches are converted into input query tokens. The VGQ module is utilized to extract vertebrae-aware contextual information by channel and spatial attention mechanisms. Then, vertebrae-related patches or tokens are highlighted by this contextual information. Thus, the VGQ module can leverage global information to locate the spine across the whole image, improving the segmentation capabilities of our VerFormer on the spine from CT images.

**Figure 4 diagnostics-14-01859-f004:**
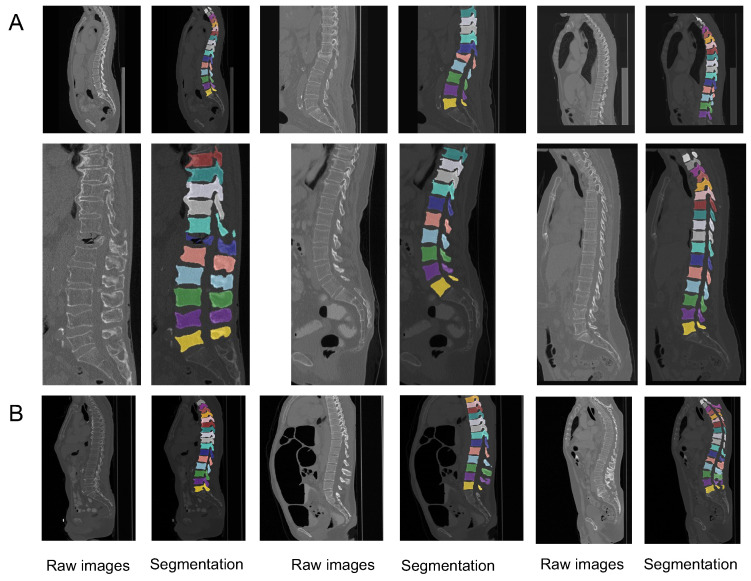
The visualization of segmentation results from our VerFormer. (**A**) The visualization of segmentation results from our VerFormer on the VerSe 2019 dataset. (**B**) The visualization of segmentation results from our VerFormer on the VerSe 2020 dataset.

**Table 1 diagnostics-14-01859-t001:** Comparison of spine segmentation between VerFormer and SOTA methods on the VerSe 2019. Segmentation performance was evaluated by mean DSC (%), standard deviation (SD) in DSC (%), IoU (%), precision (%), recall (%), and 95HD (mm). The best segmentation results are in **bold**. (*: p<0.01 with Wilcoxon signed-rank test between our V-Former and each method).

Backbones	Methods	Mean DSC	SD DSC	IoU	Precision	Recall	95HD
CNN	UNet [48]	88.14	6.01	87.21	93.56	93.11	11.54
	VNet [49]	89.51	5.45	88.03	93.28	92.21	10.63
	Att-UNet [50]	89.97	5.36	90.25	93.68	92.74	10.21
	nnU-Net [54]	93.53	3.24	94.21	98.60	97.34	8.42
CNN+Transformer	TransUNet [23]	91.58	4.75	90.88	95.76	94.89	9.88
	UCTransNet [51]	90.12	4.89	90.42	94.12	93.92	10.02
	3D UNETR [25]	94.21	3.11	95.12	98.78	97.92	5.42
	Swin UNETR [26]	94.02	3.05	94.34	97.42	97.80	5.78
	3D UX Net [52]	94.37	3.02	95.19	98.89	97.96	5.24
Transformer	Swin UNet [24]	90.25	4.66	91.42	96.15	95.85	9.98
	MissFormer [53]	93.52	3.25	92.29	97.11	96.13	8.40
	AgileFormer [31]	94.15	3.17	93.98	98.88	97.41	5.62
	VerFormer (ours)	95.28 *	2.89	95.90 *	99.57 *	98.45 *	4.24 *

**Table 2 diagnostics-14-01859-t002:** Comparison of spine segmentation between VerFormer and SOTA methods on the VerSe 2020. Segmentation performance was evaluated by mean DSC (%), standard deviation (SD) in DSC (%), IoU (%), precision (%), recall (%), and 95HD (mm). The best segmentation results are in **bold**. (*: p<0.01 with Wilcoxon signed-rank test between our V-Former and each method).

Backbones	Methods	Mean DSC	SD DSC	IoU	Precision	Recall	95HD
CNN	UNet [48]	87.98	6.27	88.45	92.62	92.41	12.56
	VNet [49]	88.11	6.32	87.24	93.45	93.04	11.62
	Att-UNet [50]	89.45	6.11	90.51	94.18	92.88	10.81
	nnUNet [54]	91.45	4.35	91.49	95.04	94.52	9.96
CNN+Transformer	TransUNet [23]	90.44	5.88	91.21	94.32	93.45	9.74
	UCTransNet [51]	90.02	5.21	90.12	93.76	93.16	10.36
	3D UNETR [25]	92.35	4.87	92.67	97.72	97.46	8.88
	Swin UNETR [26]	92.21	4.63	92.35	97.51	97.24	9.02
	3D UX Net [52]	92.73	4.52	92.75	97.81	97.66	8.62
Transformer	Swin UNet [24]	90.01	5.97	91.34	96.18	95.98	10.38
	MissFormer [53]	92.22	4.66	92.29	97.11	96.13	9.05
	AgileFormer [31]	92.64	4.14	92.80	98.02	97.22	8.74
	VerFormer (ours)	93.98 *	3.21	94.02 *	99.10 *	98.75 *	5.88 *

**Table 3 diagnostics-14-01859-t003:** Comparison of spine segmentation between VerFormer and ViT on the VerSe 2019 and VerSe 2020 datasets. Segmentation performance was evaluated by DSC (%), IoU (%), precision (%), recall (%), and 95HD (mm). The best segmentation results are in **bold**. (*: p<0.01 with Wilcoxon signed-rank test between our V-Former and ViT).

Datasets	Methods	DSC	IoU	Precision	Recall	95HD
VerSe 2019	ViT [8]	90.17	91.12	95.95	95.87	9.99
	VerFormer (ours)	95.28 *	95.90	99.57	98.45	4.24
VerSe 2020	ViT [8]	89.86	91.25	96.01	95.72	10.52
	VerFormer (ours)	93.98	94.02	99.10	98.75	5.88

## Data Availability

All data included in this study are available upon request by contact with the corresponding author.

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
