# Peer review of "VerFormer: Vertebrae-Aware Transformer for Automatic Spine Segmentation from CT Images"

_diagnostics, 2024, doi:10.3390/diagnostics14171859_

Round 1

Reviewer 1 Report

Comments and Suggestions for Authors

The proposed work targets spine segmentation in CT images, particularly the limitations of convolutional neural networks in capturing global contextual information. While the vision transformer addresses this, it lacks specificity in distinguishing vertebrae-related tokens. To alleviate this, the authors present a vertebrae-aware global block to enhance global contextual understanding and improve spine segmentation accuracy. While this is an important challenge to solve, the proposed work has a number of limitations.

1) Authors mention class imbalance - can this be explained in more detail, especially since the vertebrae still take up a large region in the images.

2) Can the authors explain how exactly do they resolve the class imbalance using the combination of dice and cross-entropy; while it shows that it helps, it is not always evident for applications such as spine segmentation, where the class imbalance is different compared to segmenting small lesions of few pixels and especially when combined with newer models like a transformer; moreover, how did the authors decide on the weighting of the losses.

3) Have the authors considered working with 3D patches, especially given the nature of CT images and the uniform spacing? CNNs employed on 3D images show a better capability to make use of the depth information - it would be interesting to see whether the proposed approach could be improved.

4) Can the authors explain the choice of data augmentation techniques? More details need to be included on how often the augmentations are performed, what level of Gaussian noise is applied (mean and standard deviation), what does “adding contrasts” mean, etc. Moreover, why did the authors not use any spatial augmentations, such as affine or elastic transforms - these are shown to be important for data variation if employed properly.

5) Why did the authors opt for cross-validation?

6) The authors report two overlap-based scores - it would be highly beneficial to report a boundary-based metric, such as the Hausdorff distance, especially in comparison to the 95th percentile of the Hausdorff distance that also outlines large errors and outliers.

7) The comparison of the proposed work in 2D to 3D methods may not be completely fair - it is strongly recommended to study the performance of the approach in 3D as well.

8) More detail on training and fine-tuning of all the other models compared should be provided. If the optimization is not optimal, the comparison could be unfair, even if all parameters are kept the same. The main issue is that there are quite substantial differences between different architectures and some may require certain parameters to be set to higher values in order to achieve optimal results.

9) The authors should provide standard deviation/variance in addition to all of the mean scores reported. Some values are too close to each other and it is not entirely evident where do the significant improvements that are claimed come from.

10) The authors should provide a visual comparison of the results with other networks tested and compared against - basically, expand Figure 4.

11) In line with the previous comment, it is not entirely clear where the improvement comes from - a deeper analysis and understanding of which vertebrae under-perform or are improved is needed. Would be also beneficial to inspect the improvement per vertebrae separately, since separate classes are already available.

12) Can the authors comment on the effects of their methods vs others on variation in resolution and size of the vertebrae? Figure 3 indicates that there might be significant differences within the data and it might actually be the case that the improvements reported in the results come from the proposed method being able to handle that (especially since it is patch-based) compared to other methods.

13) Can the authors comment on the generalization capabilities of their method? What happens if the method is trained on a dataset collected at 1 center and employed on the dataset from another center/study?

14) Section 4.1, “adding contrasts” repeated twice

Comments on the Quality of English Language

Well written, minor editing needed for repetition.

Author Response

Comment 1: Authors mention class imbalance - can this be explained in more detail, especially since the vertebrae still take up a large region in the images.

Response 1: our model is a 2D model, and it will take 2D patches as input. In some patchs, the vertebrae will only take a small region in the images. However, to avoid misunderstanding, we have removed all text related to “class imbalance” from our manuscript.

Comment 2: Can the authors explain how exactly do they resolve the class imbalance using the combination of dice and cross-entropy; while it shows that it helps, it is not always evident for applications such as spine segmentation, where the class imbalance is different compared to segmenting small lesions of few pixels and especially when combined with newer models like a transformer; moreover, how did the authors decide on the weighting of the losses.

Response 2: Dice loss is insensitive to class imbalance, so we resolved the class imbalance by seting a lager parameter for Dice loss than the cross entropy loss. As we mentioned, to avoid misunderstanding, we have removed all text related to “class imbalance” from our manuscript. We used the grid search technique to decide the weighting of the losses. We started from an initial weight for losses that were used in our paper, and then we used the grid search to find the optimal parameters in our experiment.  We have updated details about the optimal weighting in the Section 3.3 Loss Function.

Comment 3: Have the authors considered working with 3D patches, especially given the nature of CT images and the uniform spacing? CNNs employed on 3D images show a better capability to make use of the depth information - it would be interesting to see whether the proposed approach could be improved.

Comment 7 The comparison of the proposed work in 2D to 3D methods may not be completely fair - it is strongly recommended to study the performance of the approach in 3D as well.

Response 3 and 7: We appreciate these suggestions, and they are very helpful. However, our VerFormer extracts tokens from the whole image to utilize the global information. Thus, if our VerFormer takes 3D patches as input, the multi-head self-attention module will have a high computational complexity. We considered this suggestion very carefully, and decided to explain the above reason in the Section 6 Discussion in our manuscript. Additionally, although our VerFormer is a 2D model, it has superior performance than other 3D SOTA models, such as UNETR, Swin UNETR, and UX Net.

Comment 4: Can the authors explain the choice of data augmentation techniques? More details need to be included on how often the augmentations are performed, what level of Gaussian noise is applied (mean and standard deviation), what does “adding contrasts” mean, etc. Moreover, why did the authors not use any spatial augmentations, such as affine or elastic transforms - these are shown to be important for data variation if employed properly.

Response 4: We thank the comment. We have included all details about data augmentation techniques in the Section 4.1 Datasets in our manuscript.

Comment 5: Why did the authors opt for cross-validation?

Response 5: We used the cross-validation to avoid overfitting and to provide a more accurate estimate of the model’s generalization performance. 5-fold cross validation is widely used in experiments of medical image segmentation. We included this explanation in our manuscript.

Comment 6: The authors report two overlap-based scores - it would be highly beneficial to report a boundary-based metric, such as the Hausdorff distance, especially in comparison to the 95th percentile of the Hausdorff distance that also outlines large errors and outliers.

Response 6: We appreciate this comment. We reported 95% Hausdorff distance (95HD) in our experimental results.

Comment 8: More detail on training and fine-tuning of all the other models compared should be provided. If the optimization is not optimal, the comparison could be unfair, even if all parameters are kept the same. The main issue is that there are quite substantial differences between different architectures and some may require certain parameters to be set to higher values in order to achieve optimal results.

Response 8: These methods were not applied to these two spine segmentation datasets, so we used the default training and fine-tuning parameters of all other models that were mentioned in their published paper.

Comment 9: The authors should provide standard deviation/variance in addition to all of the mean scores reported. Some values are too close to each other and it is not entirely evident where do the significant improvements that are claimed come from.

Response 9: Thanks for this suggestion. We have included standard deviation of Dice Score (DSC) in experimental results from two datasets. Other metrics, such as IoU, precision, and recall, can be calculated and inferred from the DSC, so we only reported the standard deviation of DSC.

Comment 10: The authors should provide a visual comparison of the results with other networks tested and compared against - basically, expand Figure 4.

Response 10: We tried to provide a visual comparison before, but as mentioned in the Comment 9, other models had close segmentation values. Thus, the visual comparison is not very significant. However, to demonstrate that the improvements of our model over other methods were significant, we implemented the Wilcoxon signed-rank test between our V-Former and each method. Thus, we expect we can show significant improvements.

Comment 11: In line with the previous comment, it is not entirely clear where the improvement comes from - a deeper analysis and understanding of which vertebrae under-perform or are improved is needed. Would be also beneficial to inspect the improvement per vertebrae separately, since separate classes are already available.

Response 11: Thanks for the suggestion. However, it is not easy to analyze which vertebrae are improved. Our VerFormer is designed as a general segmentation model, and it utilizes the global contextual information to highlight the whole spine region, thus improving the overall segmentation performance. It takes the whole image as the input, and employs a 1x1 convolutional layer to generate voxel-wise prediction. It does not employ any techniques to extract features of any specific vertebrae or to refine the segmentation result of any specific vertebrae. Additionally, we have evaluated the improvements of our Vertebrea-aware Global (VG) block in spine segmentation in the Section 5 Ablation Study. In this section, we investigated that the improvements come from the incorporation of the VG block in our VerFormer.

Comment 12: Can the authors comment on the effects of their methods vs others on variation in resolution and size of the vertebrae? Figure 3 indicates that there might be significant differences within the data and it might actually be the case that the improvements reported in the results come from the proposed method being able to handle that (especially since it is patch-based) compared to other methods.

Response 12: We use the same patch size and resolution as other models. We used the sliding window inference to make predictions, so the patch size in our model are same as it in other models. Variation in resolution and size will not influence the final results.

Comment 13: Can the authors comment on the generalization capabilities of their method? What happens if the method is trained on a dataset collected at 1 center and employed on the dataset from another center/study?

Response 13: We appreciate this suggestion. We commented the generalization capabilities of our method in the Section 6 Discussion.

Comment 14: Section 4.1, “adding contrasts” repeated twice

Response 14: Thanks for the comment. We have corrected this typo.

Reviewer 2 Report

Comments and Suggestions for Authors

The manuscript presents a significant advancement in automatic spine segmentation using a novel Vision Transformer-based approach. The proposed model, which integrates a Vertebrae-aware Global (VG) block into the Vision Transformer backbone, demonstrates impressive improvements in segmentation accuracy. The introduction effectively highlights the challenges of spine segmentation and the limitations of traditional convolutional neural networks (CNNs), providing a solid rationale for the proposed method. However, some areas could benefit from further refinement.

While comprehensive, the literature review would be strengthened by including more recent studies exploring the application of other AI methods in different areas of medical imaging (10.3390/diagnostics13172760, 10.1109/ISBI53787.2023.10230448). This would better contextualize the novelty of the approach and illustrate how it builds on existing research (10.1016/j.ejrad.2023.110786, 10.26044/ecr2023/C-16014). Additionally, a more detailed discussion of related work would help differentiate contributions from previous studies' contributions.

The methodology section is well-detailed and provides clear explanations of the innovative components of the model, such as the Vertebrae-aware Global Query (VGQ) module. Including more specifics about the datasets used, including any preprocessing steps and the rationale behind confident parameter choices (10.7759%2Fcureus.41435, 10.1109/ISBI53787.2023.10230686), would enhance the transparency and reproducibility of the work. The experimental setup is robust, but a more thorough explanation of the evaluation metrics would benefit readers less familiar with these techniques.

The results are compelling, showing improvements over existing methods in both the VerSe 2019 and VerSe 2020 datasets. The use of visual aids, such as segmentation result comparisons, effectively illustrates the performance of the model. Providing additional examples of successes and failures could offer a more nuanced understanding of the model’s performance and potential areas for improvement.

The conclusion briefly summarizes the findings and emphasizes the significance of the contributions to the field. Including a brief discussion of future research directions or potential applications of the work would be valuable, as it could inspire further innovation and exploration in this area.

While the quality of English is generally good, there are a few areas where minor editing could improve clarity and readability. Simplifying complex sentences and ensuring consistent use of terminology throughout the manuscript would enhance the overall flow of the text.

In summary, this manuscript presents a highly relevant and innovative approach to spine segmentation. With some enhancements to the literature review and additional methodological details, it would make a solid contribution to the field. The suggestions provided should help improve the clarity and impact of the work.

Comments on the Quality of English Language

The quality of English in the manuscript is generally reasonable. However, a few areas could benefit from minor editing to improve clarity and readability. Simplifying complex sentences and ensuring consistent use of terminology throughout the manuscript would enhance the overall flow of the text. Additionally, addressing minor grammatical issues and improving sentence structure in some sections would further refine the manuscript. These improvements will help to convey the innovative contributions of the research to a broader audience.

Author Response

Comment 1: While comprehensive, the literature review would be strengthened by including more recent studies exploring the application of other AI methods in different areas of medical imaging (10.3390/diagnostics13172760, 10.1109/ISBI53787.2023.10230448). This would better contextualize the novelty of the approach and illustrate how it builds on existing research (10.1016/j.ejrad.2023.110786, 10.26044/ecr2023/C-16014). Additionally, a more detailed discussion of related work would help differentiate contributions from previous studies' contributions.

Response 1: We appreciate this comment. We included more literatures and papers related to Vision Transformer and Spine Segmentation in the Section 1 Introduction and Section 2 Related Work. Some studies that were mentioned below are about breast cancer detection. We did not include them since they were not related to our paper. In our manuscript, we have included detailed discussion of work related to out work, such as Convolutional neural network in spine segmentation, Vision transformer in medical image segmentation, and attention mechanism.

Comment 2: The methodology section is well-detailed and provides clear explanations of the innovative components of the model, such as the Vertebrae-aware Global Query (VGQ) module. Including more specifics about the datasets used, including any preprocessing steps and the rationale behind confident parameter choices (10.7759%2Fcureus.41435, 10.1109/ISBI53787.2023.10230686), would enhance the transparency and reproducibility of the work. The experimental setup is robust, but a more thorough explanation of the evaluation metrics would benefit readers less familiar with these techniques.

Response 2: We thank reviewer’s comment. We have included more details about out preprocessing and data augmentation steps in the Section 4.1 Datasets. We also included more details about the hyperparameters of our network Section 3.1 Overall Network to enhance the transparency and reproducibility of the work.

Comment 3: The results are compelling, showing improvements over existing methods in both the VerSe 2019 and VerSe 2020 datasets. The use of visual aids, such as segmentation result comparisons, effectively illustrates the performance of the model. Providing additional examples of successes and failures could offer a more nuanced understanding of the model’s performance and potential areas for improvement.

Response 3: We agree showing some examples of successes will have benefits, so we have shown some good segmentation results in Figure 4. On the contrary, our model has a very great segmentation results, over 95%, with a low standard deviation. Thus, it is not easy to find an example of failures to provide a more nuanced understanding. To present the improvements of our model over other models, we used 5 different metrics to show the results. We also implemented the Wilcoxon signed-rank test between our V-Former and each method to demonstrate that these improvements were significant (p<0.01). Thus, we expect we can show our results thoughtfully.

Comment 4: The conclusion briefly summarizes the findings and emphasizes the significance of the contributions to the field. Including a brief discussion of future research directions or potential applications of the work would be valuable, as it could inspire further innovation and exploration in this area.

Response 4: We appreciate this suggestion, and it is very helpful. We have added a new Section 6 Discussion. In this section, we discussed the generalizability, limitations of our models, and its future direction and improvements.

Comment 5: While the quality of English is generally good, there are a few areas where minor editing could improve clarity and readability. Simplifying complex sentences and ensuring consistent use of terminology throughout the manuscript would enhance the overall flow of the text.

Response 5: We thank the comment. We have refined our manuscript by simplifying some complex sentences and checking consistent use of terminology.